# “*A Tale of Two Cities*”: Anticoagulation Management in Patients with Atrial Fibrillation and Prosthetic Valves in the Era of Direct Oral Anticoagulants

**DOI:** 10.3390/medicina55080437

**Published:** 2019-08-04

**Authors:** Giuseppe Palmiero, Enrico Melillo, Antonino Salvatore Rubino

**Affiliations:** 1Department of Cardiology, AORN Ospedali dei Colli-Monaldi Hospital, 80131 Naples, Italy; 2Department of Translational Medical Sciences, University of Campania “Luigi Vanvitelli”, 80131 Naples, Italy

**Keywords:** novel oral anticoagulants, prosthetic valve replacement, atrial fibrillation

## Abstract

Valvular heart disease and atrial fibrillation often coexist. Oral vitamin K antagonists have represented the main anticoagulation management for antithrombotic prevention in this setting for decades. Novel direct oral anticoagulants (DOACs) are a new class of drugs and currently, due to their well-established efficacy and security, they represent the main therapeutic option in non-valvular atrial fibrillation. Some new evidences are exploring the role of DOACs in patients with valvular atrial fibrillation (mechanical and biological prosthetic valves). In this review we explore the data available in the medical literature to establish the actual role of DOACs in patients with valvular heart disease and atrial fibrillation.

## 1. Prologue

In 1859 Charles Dickens wrote “*A tale of two cities*”, a historical novel set in London and Paris during the French Revolution, a transition period from the monarchical ancient regime to a new form of government. In medicine, now as then, with the introduction into the clinical practice of new therapeutic options for anticoagulation management, a new era is rising, but contrasts are still present. Even in fields of application in which new anticoagulants have proved to be superior to the old molecules (thrombosis prevention in atrial fibrillation and systemic thromboembolism), a sense of comfortability, given by decades of clinical experience of historical anticoagulant drugs, is still contrasting the potential widespread use of new therapeutic option, encouraged by their favorable outcome in terms of efficacy and security.

Moreover, the ancient regime of oral vitamin K antagonists (VKAs) is still dominating the anticoagulation management in prosthetic valve replacement. However, some new evidence seems able to promote a possible revolution in this setting.

## 2. Introduction

Prosthetic valve replacement represents a historical and effective therapeutic approach for symptoms and outcome improvement in patients with valvular heart disease (VHD) [1]. However, its effectiveness is counterbalanced by complications, whose frequency and severity depend upon valve type and position, as well as other patient-specific risk factors. Thromboembolic events are among those potential prosthesis-related complications, especially early after valve implantation [2].

Biological prosthetic valves, mechanical prosthetic valves, and more recently transcatheter biological prosthetic valve represent the options available in case of natively diseased or damaged heart valves. The decision about the use of one of these prosthetic valves depends upon patient’s age, a punctual balance between thromboembolic and bleeding factors, and patient’s decision [3].

Long-term anticoagulation therapy has been based historically on oral VKAs [4]. Those agents have been used for some time, accumulating through these decades robust data and clinical experience. For these reasons, despite numerous important limitations, physicians are still confident with their use. Novel direct oral anticoagulants are target-selective agents, including direct thrombin, or factor IIa (dabigatran), and factor Xa inhibitors (apixaban, rivaroxaban, edoxaban). In meta-analysis, including patients across all registration trials, those agents have shown superiority to warfarin (the main VKA agent) for the prevention of stroke and systemic embolism in non-valvular atrial fibrillation [5,6]. DOACs have several pharmacokinetic advantages compared to VKAs: A fixed dose, which avoids the necessity for drug monitoring, rapid onset of action and a short-half life, which limits their action during a short time, few interactions with food and drugs, which makes their use easier and safer. Those characteristics make the DOACs an attractive to VKAs in long-term anticoagulation management of patients with prosthetic valves. However, currently, data showing a net benefit of DOACs in anticoagulation management of prosthetic valve are still lacking. 

Recently, in a European Heart Rhythm Association (EHRA) position paper [7] about antithrombotic therapy in atrial fibrillation (AF), a new categorization of VHD in relation to the type of oral anticoagulation use has been proposed. Considering the terms “valvular” and “non-valvular” AF as outdated, the functional EHRA (evaluated heart valves rheumatic or artificial) categorization proposes a distinction in type 1 form, which refers to AF patients with VHD needing therapy with VKAs (moderate-to-severe mitral stenosis of rheumatic origin), and type 2 form, which refers to AF patients with VHD needing therapy with VKAs or DOACs (heart valve regurgitation, aortic stenosis, tricuspid stenosis, pulmonic stenosis, mild mitral stenosis, mitral valve repair, transaortic valve intervention, and bioprosthetic valve replacement). However, this categorization is in contrast with the purpose of the review, namely exploring the possible use of DOACs even in the EHRA type 1 forms, in which these agents are currently contraindicated.

DOACs in AF have showed to but safe and effective in different scenarios: in the elderly [8], in cancer patients [9,10], and in real-life experiences [11,12]. Moreover, DOACs have showed to be well tolerated, with low rates of discontinuation, in real-life [13] as in pivotal trials. Periprocedural and long-time anticoagulation therapy management in patients with AF undergoing interventional procedures represents one of the great chapters related to the use of anticoagulants. Numerous clinical trials have investigated the use of individual drugs in various scenarios and have been translated in recent joint consensus documents among various scientific societies (antithrombotic management in transcatheter valve replacement or repairment [14], percutaneous coronary interventions [15,16], electrophysiological procedures [17], and AF cardioversion [18]). Recent data, analyzing the clinical performance of DOACs in real-world patients who underwent interventional procedures, are emerging. DOACs have shown to be, also in real-world data, both effective and safe in many scenarios (for example, in electrical AF cardioversion [19], also in a population at a high thromboembolic and hemorrhagic risk [20]), confirming the data emerged from dedicated trials. Conversely, real-world data about anticoagulation therapy management in patients undergoing surgical or percutaneous valvular interventions are still missing.

## 3. Discussion

### 3.1. Anticoagulation Therapy in Prosthetic Valves

#### 3.1.1. Anticoagulation Therapy in Mechanical Prosthetic Valves

Mechanical prosthetic valves (MPV) are often chosen in younger patients for their intrinsic high durability and low incidence of valve failure. However, these advantages are counterbalanced by a significant increase in thromboembolic events, especially shortly after valve implantation. Multiple thrombotic risk factors are concomitantly present in the vast majority of patients (older age, hypercoagulable state, a history of congestive heart failure, chronic kidney disease or atrial fibrillation, etc.) and for these reasons, long-term anticoagulation is mandatory for MPV and the VKAs are the current standard of care in these patients [3,21].

#### 3.1.2. Anticoagulation Therapy in Biological Prosthetic Valves

Biological prosthetic valves (BPV) are considered less thrombogenic than MPV. However, valve thrombosis in the absence of anticoagulation therapy should not be underestimated, especially in the presence of known risk factors, such as low cardiac output state and structural valve deterioration [2]. Therefore, in patients in sinus rhythm with BPV, following current ACC/AHA guidelines recommendation [22], short-term anticoagulation and antithrombotic management with VKA and low-dose aspirin should be co-administrated for the three to six months after valve implantation. On the other side, ESC/EACTS 2017 guidelines recommend only a short-term (three months) anticoagulation management with VKA after mitral bioprosthetic implantation or surgical valvuloplasty. However, as bioprosthetic aortic valves are considered less thrombogenic than mitral ones [14], short-term antithrombotic management with low-dose aspirin is preferred to anticoagulation therapy. On the other hand, in patients with AF and BPV, long-term anticoagulation with VKA is mandatory, and the general increase in life expectancy is leading to a more frequent association between these two. The thromboembolic risk, in this setting, may be related to both BPV and to AF. However, the incidence of thromboembolic events is similar to those of age-matched patients with chronic AF only [23].

#### 3.1.3. Anticoagulation Therapy After Transcatheter Aortic Valve Implantation

Transcatheter bioprosthetic aortic implantation (TAVI) represents a new therapeutic option for patients with symptomatic severe aortic valve stenosis considered at moderate-to-high risk of surgical replacement. The ESC/EACTS guidelines [3] suggest a dual antiplatelet therapy for the first three to six months after TAVI, followed by lifelong single antiplatelet therapy. In case of high bleeding risk, the dual antiplatelet therapy should be avoided, and the lifelong antithrombotic management should be based on a single antiplatelet therapy. However, new evidence has shown that those valves determine a higher risk of subclinical leaflet thrombosis than surgical prostheses, without significant differences in terms of incidence of stroke [24]. Considering this, current ACC/AHA guidelines [21] have suggested anticoagulation management (non-fractioned heparin for the time interval needed to achieve a 2.5 target INR with VKA) for at least three-months after TAVI, in the absence of a high bleeding risk.

### 3.2. Role of Direct Oral Anticoagulants in Prosthetic Valves

#### 3.2.1. DOACs in Patients with Atrial Fibrillation and Biological Prosthetic Valves

In AF alone, DOACs represent an effective and safe alternative to VKA. Due to their attractive pharmacokinetic profile, their use has been recently extended to patients with MPV and AF despite the lack of prospective controlled data. Indeed, DOACs have only been restricted for cases with “non-valvular” AF in the currently available trials, and only very few patients with mitral bioprosthesis have been enrolled on the ARISTOTLE [25] and ENGAGE-AF-TIMI48 trials [26]. 

Guimaraes at al. [27] have recently explored the efficacy and safety of apixaban versus warfarin in patients with AF and prior BPV replacement or valve repair, analyzing the data obtained from patients enrolled in the apixaban pivotal trial (ARISTOTLE). Of more than 18,000 patients enrolled, only 0.6% (104 patients, n = 76 aortic, n = 23 mitral and n = 5 aortic and mitral) had a history of BPV replacement: 55 were randomized to apixaban and 49 to warfarin. Moreover, about 0.3% had a history of valve repair (52 pts, n = 50 mitral and n = 2 aortic): 32 were randomized to apixaban and 20 to warfarin. Efficacy outcomes included stroke or systemic embolism, all-cause stroke, ischemic stroke, myocardial infarction, all-cause death, and cardiovascular death. Safety outcomes included major bleeding, major or clinically evident non-major bleeding, intracranial hemorrhage, gastrointestinal bleeding, and any bleeding. In this subgroup analysis, no significant differences were found between the groups for any of the characteristics analyzed, showing that apixaban is safe and effective also in patients with AF and prior BPV replacement or valve repair. Those results were consistent with results shown in the main ARISTOTLE pivotal trial. 

In the edoxaban pivotal ENGAGE AF-TIMI 48 trial patients with left-sided valvular heart disease were enrolled, including those with a history of aortic or mitral valve surgical or transcatheter implantation more than 30 days before randomization [28]. Of more than 21,000 patients enrolled, 0.9% had a previous BPV replacement (191 pts, n = 131 mitral, n = 60 aortic), and among them, 70 patients were randomized to warfarin, 63 patients to high-dose of edoxaban (60 mg daily) and 58 to low-dose of edoxaban (30 mg daily). Primary endpoints included stroke and systemic embolic events, major bleeding, and the primary net clinical outcome. Secondary composite endpoints included ischemic stroke, major adverse cardiac events (myocardial infarction, stroke, or cardiovascular death), and the composite of stroke, all-cause mortality, and life-threatening bleeding. In a subgroup analysis, patients with BPV treated with higher dose edoxaban had similar rates of stroke and major bleeding and lower rates of cardiovascular events (myocardial infarction, cardiovascular death) and primary net clinical outcome compared with warfarin. Patients treated with lower dose edoxaban had similar rates of stroke but lower rates of major bleeding and of the primary net clinical outcome compared with warfarin. In this analysis, edoxaban appears to be a reasonable alternative to warfarin in patients with AF and previous BPV implantation. 

However, both the sub-analyses of DOAC pivotal trials have important limitations, including a small sample size and low number of events. Then, to definitively establish the role for alterative anticoagulation to VKA in this setting, larger dedicated controlled trials are needed to definitively assess the safety and efficacy of DOACs. 

Russo et al. [29] have proposed in 2018 a multicenter observational study to investigate the efficacy and safety of DOACs in AF patients with BPV or prior surgical valve repair. A total of 122 patients were enrolled. In 92% of cases, warfarin was replaced due to lack of compliance and subtherapeutic INR range. The study population included 24 patients (19.6%) with mitral BPV, 52 patients (43%) with aortic BPV, 41 patients (33.6%) with previous surgical mitral repair, and 5 patients (4%) with a previous surgical aortic repair. Of the total study population, 28.6% were taking apixaban 5 mg twice daily, 24.5% apixaban 2.5 mg twice daily, 18% dabigatran 150 mg twice daily, 13% dabigatran 110 mg twice daily, 9.8% rivaroxaban 20 mg daily, and 5.7% rivaroxaban 15 mg daily. All patients were evaluated for thromboembolic events (ischemic stroke, transient ischemic attack, systemic embolism) as well as major bleeding events during the follow-up period and showed a low mean annual incidence of thromboembolism (0.8%) and major bleeding (1.3%). According to this data, DOAC therapy seems to be an effective and safe treatment alternative for AF patients with BPV or prior surgical valve repair. However, this study is limited by small sample size, a retrospective design, heterogeneous anticoagulation management and lack of VKA control group.

At present, the use of DOACs for the management of concomitant AF following BPV replacement may be considered a valid therapeutic option, with the exception of biological mitral prosthesis implanted in the setting of rheumatic mitral stenosis [30]. However, dedicated double-blinded trials confronting DOACs to VKA in this setting are necessary to evaluate the actual efficacy and safety of recommending the DOACs in patients with bioprosthetic valves.

#### 3.2.2. DOACs in Patients with Atrial Fibrillation and Mechanical Prosthetic Valves

The only published trial investigating the use of a DOACs in mechanical prosthetic valves is the RE-ALIGN Trial [31]. The study made a comparison between dabigatran and warfarin in patients with aortic valve replacement with a mechanical prosthesis for the prophylaxis of thromboembolic events. The study was stopped early after enrollment of only 252 patients due to a significant increase in both thromboembolic (5% in dabigatran group vs. 0% in warfarin group) and bleeding (4% in dabigatran group vs. 2% in warfarin group) events in patients treated with dabigatran compared to conventional therapy with warfarin. Therefore, the use of DOACs in this particular subset of patients appeared of no benefit and excessively harmful, so that the trial was prematurely discontinued. Different mechanisms of action of dabigatran and warfarin can partially explain these findings. In patients with mechanical prosthesis, thrombus formation can derive from contact pathway of coagulation, triggered by exposure of blood to the artificial elements of the valve (ring, struts, leaflets) and from direct release of prothrombotic tissue factor from damaged tissues during surgery. Therefore, VKAs are more effective than dabigatran in this setting by stopping both tissue factor and contact pathway induced coagulation [32]. 

Despite discouraging results from the RE-ALIGN Trial, Durães and coworkers [33] designed a pilot study to investigate the potential role of rivaroxaban as an alternative to VKA in patients with mechanical prosthetic valves. Their rationale [34,35] was that clotting on the mechanical valves is triggered by the contact and that dabigatran administered in the RE-ALIGN Trial was insufficient to inhibit thrombus formation in this scenario. On the other hand, rivaroxaban is a direct inhibitor of Factor Xa with potential to reduce significantly the generation of thrombin on mechanical prostheses. Accordingly, seven patients with mechanical mitral prosthesis received rivaroxaban 15 mg twice daily and were followed-up for 90 days. At the end of the study, no patients experienced neither thromboembolic nor bleeding adverse events. These findings opened the way to a subsequent randomized controlled trial, whose enrolling phase is expected to end in December 2019.

#### 3.2.3. DOACs After Transcatheter Aortic Valve Implantation in Atrial Fibrillation Patients

The use of DOACs after TAVI has been investigated in the GALILEO trial [36]. In the study, rivaroxaban (10 mg once daily) plus short-term (three months) antiplatelet therapy with low dose aspirin (75 to 100 mg once daily) has been compared to short-term dual antiplatelet therapy management (clopidogrel 75 mg plus aspirin 75 to 100 mg once daily for three months), followed by long-term single antiplatelet management with aspirin alone for thromboembolic prevention after TAVI. The primary efficacy endpoint of the trial is a composite of all-cause death, stroke, systemic embolism, MI, pulmonary embolism, deep vein thrombosis, or symptomatic valve thrombosis. The primary safety endpoint is a composite of life-threatening or disabling bleeding or major bleeding. The trial has been halted precociously: From preliminary data released by Bayer, the rivaroxaban-based antithrombotic strategy has shown an increase in the rates of death or first thromboembolic event (11.4% vs. 8.8%), all-cause death (6.8% vs. 3.3%), and primary bleeding (4.2% vs. 2.4%) compared to antiplatelet-based therapy. Therefore, this confirmed that the use of DOACs is contraindicated after TAVI.

A summary of the main clinical trials and subgroup analysis is reported in Table 1.

## 4. Conclusions

As for all huge transformations occurring in the human history, there always exists a contrast between the forces of changes, which promise to revolutionize the previous status quo, and the forces of reaction, which react against them with the certainties accumulated over the years. 

The revolution promised by the novel direct oral anticoagulants in the management of anticoagulant therapy in patients undergoing valve replacement is likely to remain inconclusive: The few clinical trials comparing DOACs to warfarin have been shown to increase the risk of all-cause of mortality, thromboembolic events, and bleeding in patients with MPV. Moreover, a single clinical trial with a single DOAC does not represent robust and clear evidence for dismissing a therapeutic strategy. Furthermore, we should consider that not all DOACs are equal in terms of effectiveness for different indications. Therefore, long-term anticoagulation therapy with VKAs is still mandatory in the setting of mechanical valve replacement.

Even after transcatheter aortic valve replacement, DOACs did not show an advantage over warfarin in terms of thromboembolic bleeding risk reduction. However, the concomitant presence of both thromboembolic and hemorrhagic risk factors, and the lack of robust data, make it difficult to establish the best antithrombotic strategy in this setting.

In patients with BPV, some observations obtained by sub-analysis of pivotal trials data are encouraging. However, in the absence of dedicated double-blinded trials confronting DOACs to VKA in terms of efficacy and safety, there is little evidence of treatment with DOACs in clinical practice in patients with bioprosthetic valves.

## Figures and Tables

**Table 1 medicina-55-00437-t001:** Summary of the main clinical trials and subgroup analysis assessing the clinical performance of novel oral anticoagulants in patients with bioprosthetic or mechanical heart valve.

Clinical Trial	Prosthetic Valve	N. Patients	Anticoagulant Regimen	Primary Efficacy Endpoint	Safety Endpoint
ARISTOTLE	Biological	104 bioprosthetic valve	Apixaban (n = 55) Warfarin (n = 49)	No significant difference	No significant difference
ENGAGE AF-TIMI 48	Biological	191 bioprosthetic valve	Edoxaban 60 mg (n = 63) Edoxaban 30 mg (n = 58) Warfarin (n = 70)	No significant difference for stroke/systemic embolic events for edoxaban 60 mg (*P* = 0.15) and edoxaban 30 mg (*P* = 0.31) vs. warfarin. Lower rates of primary net clinical outcome with edoxaban 60 mg (*P* = 0.03) and edoxaban 30 mg (*P* = 0.03)	No significant difference between high dose edoxaban and warfarin (*P* = 0.26) Lower major bleeding events with edoxaban 30 mg (*P* = 0.045) vs. warfarin
RE-ALIGN	Mechanical	252 (trial was prematurely stopped)	Dabigatran (150–220–300 mg based on kidney function) Warfarin	9 stroke events (5%) in the dabigatran group No stroke event in the warfarin group	7 major bleeding events (4%) in the dabigatran group 2 major bleeding events (%) in the warfarin group
GALILEO	Transcatheter aortic valve replacement	1644 (trial was prematurely stopped)	Rivaroxaban 10 mg + Aspirin 100 for 90 days after TAVR Clopidogrel 75 mg + Aspirin 100 mg for 90 days after TAVR	Death and first thromboembolic events: 11.4% rivaroxaban group vs. 8.8% in antiplatelet group. All-cause death: 6.6% in rivaroxaban group vs. 3.3% in antiplatelet group.	Primary bleeding: 4.2% in rivaroxaban group vs. 2.4% in antiplatelet group.

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
