# Peer review of "A Tale of Two Cities”: Anticoagulation Management in Patients with Atrial Fibrillation and Prosthetic Valves in the Era of Direct Oral Anticoagulants"

_1010-660X, 2019, doi:10.3390/medicina55080437_

Round 1

Reviewer 1 Report

Overall, this is a well written manuscript presenting available data on the effectiveness of NOACs in heart valve disease.

Comments

1. It was made clear by the authors that currently several limitations exist in available trials. A major conclusion is potentially that a single trial with a single NOAC does not represent sufficient evidence for dismissing a therapeutic strategy. Additionally, it appears that not all NOACs are equal in terms of effectiveness for different indications.

2. If consistent with the journal instructions for authors, a Table summarizing the results would be really helpful to the readers.

3. Several typos exist in the manuscript:

in abstract, “evidence is exploring”

line 21, “a historical”

line 31, “new evidence”

line 34, “a historical”

line 40, “represent”

line 54, “characteristics make”

line 69, “underestimated”

line 74, “guidelines”

line 86, “suggest”

line 89, “new evidence has”

line 90, “determine a higher”

line 97, “represent”

line 115, “results”

line 131, “NOAC”

line 137, “lack of compliance”, meaning of labile?

line 159, delete of

line 182, “has showed”

line 248, please add year in the cited reference

Author Response

Dear Reviewer,

Thank you for careful revision of our manuscript and for your interesting and helpful remarks.

We are pleased to provide our point-by-point response to the received comments. The changes within the revised manuscript are highlighted with Track changes version of Microsoft Word and with bold red color text.

Issue n.1: "It was made clear by the authors that currently several limitations exist in available trials. A major conclusion is potentially that a single trial with a single NOAC does not represent sufficient evidence for dismissing a therapeutic strategy. Additionally, it appears that not all NOACs are equal in terms of effectiveness for different indications".

We thank you for this comment and agree with this conclusion. Therefore we modified our conclusions following reviewer's suggestion (lines 235-238).

Issue n.2: "If consistent with the journal instructions for authors, a Table summarizing the results would be really helpful to the readers".

We totally agree with this comment. Therefore we added a table (Table n.1) summarizing the main clinical trials exploring the clinical performance of NOACs in prosthetic valves.

Issue n.3: "Several typos exist in the manuscript".

We are greatful for this remark and subsequently modified the text according to this suggestion.

Reviewer 2 Report

This is a good review for understanding the current standing point of NOAC in patients after valvular surgery repair. I have no major concerned comments, however, few issues to be addressed as follows:

1) Although American guidelines do not recommend NOAC in patients with biological heart valves or after valve repair, a recent European practical guideline recommends a possibility of use of DOAC for those patients, even after TAVI more favorably. Please see and refer the paper (Europace. 2018;20:1231-1242.) in the text.

2) Please describe a possible explanation for negative impact of NOAC in patients with mechanical prosthetic valves. For example, please see a paper: Eur Heart J. 2014;35:3328-35.

3) In the reference #16, the published year was inadequately presented.

Author Response

Dear Reviewer,

Thank you for careful revision of our manuscript and for your interesting and helpful remarks.

We are pleased to provide our point-by-point response to the received comments. The changes within the revised manuscript are highlighted with Track changes version of Microsoft Word and with bold red color text.

Issue #1: “Although American guidelines do not recommend NOAC in patients with biological heart valves or after valve repair, a recent European practical guideline recommends a possibility of use of DOAC for those patients, even after TAVI more favorably. Please see and refer the paper (Europace. 2018;20:1231-1242.) in the text”.

We thank you for this comment. We modified the text including reviewer's suggestion in lines 170-172 and added reference number 16.

Issue #2: "Please describe a possible explanation for negative impact of NOAC in patients with mechanical prosthetic valves. For example, please see a paper: Eur Heart J. 2014;35:3328-35".

We are grateful to the reviewer for this remark. We provided a possible explanation in lines 184-189 and added reference number 18.

Issue #3: " In the reference #16, the published year was inadequately presented".

We thank you for this remark and modified the year of the reference.

Reviewer 3 Report

I would first congratulate the authors for their interesting review article. I have some suggestions of technical character about the current manuscript:

1.       Line 3 in the title:  “…for prosthetic valves in the era…” sounds slightly better

2.       Line 34: “…an historical” should be “…a historical”

3.       Line 51-52: “a fix dose” should be “a fixed dose”; “…which avoid the needing” – “…which avoids the necessity…”

4.       Line 69: “understimated1” is probably “underestimated”

5.       Line 73: “…administrated together for the three to six months…”  -  it sounds better, if modified like “…co-administered for three to six months…”

6.       Line 74: “guidelines3” should be “guidelines” and “recommends” – “recommend”

7.       Line 86: “suggests” should be “suggest”

8.       Line 90:  “determines an higher risk” should be “determine higher risk”

9.       Line 98-99: “…their use is not recommended due to the lack of prospective controlled data in patient with both BPV and AF”. It might be restructured to “their use is not recommended in patients with BPV and AF due to the lack of prospective controlled data.”

10.   Line 144: “event” should be “events

11.   Line 182: “…have showed to increase the rates of…”  should be  “…have shown increase in the rates of…”. The same in line 192

Author Response

Dear Reviewer,

Thank you for careful revision of our manuscript and for your interesting and helpful remarks.

We are pleased to provide our point-by-point response to the received comments. The changes within the revised manuscript are highlighted with Track changes version of Microsoft Word and with bold red color text.

Issue #1: “ I would first congratulate the authors for their interesting review article. I have some suggestions of technical character about the current manuscript".

We sincerely thank you for your opinion of the manuscript. We modified the text following your remarks.